# The inverse problem for kernel means

## Abstract

We discuss the inverse problem for the kernel embedding of measures. We identify which elements of a reproducing kernel Hilbert space which are in the cone generated by some set of kernel functions as polar dual of the Herglotz-type functions, the functions with positive imaginary part. Over certain spaces, such as Sobelev spaces, the duality to Herglotz functions reduces to a classical multivariate moment problem, and, over analytic spaces, we see more complex analytic type conditions. We give conditions for when Herglotz functions have representations in terms of kernel functions in terms of reflexive reproducing kernel Hilbert spaces. We identify the orbits of a dynamical system in terms of the Koopmanism philosophy: we give a way to decide when there is an orbit contained in some compact subset of the domain.

## 1 Introduction

Kernel methods offer powerful techniques for dealing with complex and high-dimensional data by implicitly mapping the data into a high-dimensional feature space using kernel functions, see, for example, Schölkopf et al. (1998); Muandet et al. (2017); Sriperumbudur et al. (2011). The kernel embedding of measures, also known as the kernel mean, allows us to represent probability distributions as functions in some reproducing kernel Hilbert space. The kernel embedding of measures has proven to be a useful tool for various tasks in machine learning, such as distribution comparison in Gretton et al. (2012), generative modeling in Li et al. (2015), and density estimation in Liu et al. (2016), due to its ability to leverage the Euclidean geometry of reproducing kernel Hilbert spaces.

Let $\Omega$ be a set. A **reproducing kernel Hilbert space** is a Hilbert space $\mathcal{H}$ of functions $f : \Omega \to \mathbb{C}$ such that point evaluation is a bounded linear functional. In reproducing kernel Hilbert spaces, for each $\omega \in \Omega$ there is a **kernel function** $k_\omega \in \mathcal{H}$ such that $\langle f, k_\omega \rangle = f(\omega)$. (Note that the kernel functions thus induce a natural metrizable topology on $\Omega$ which we will assume is Hausdorff for ease of discussion.) For more information on the basic theory of reproducing kernel Hilbert spaces, see Paulsen & Raghupathi (2016); Berlinet & Thomas-Agnan (2011).

As aforementioned, an apparently somewhat useful technique is various contexts is to consider the **the kernel embedding of measures** (or **kernel mean** in the case of distributions) taking a measure $\mu$ to an element $\iota(\mu)$, defined by the map

$$\iota(\mu)(z) = \int k_\omega(z) \mathrm{d}\mu(\omega).$$

Note that

$$\langle f, \iota(\mu) \rangle = \int f(\omega) \mathrm{d}\mu(\omega).$$

That is, $\iota(\mu)$ behaves like integration against $\mu$ as a linear functional on $\mathcal{H}$. See Muandet et al. (2017); Sriperumbudur et al. (2011) for a comprehensive review establishing theoretical foundations and exploring applications. The utility of the kernel embeddings of measures has proven significant recently for its ability to represent probability distributions in a reproducing kernel Hilbert space, which in principle may be more desirable to manipulate, calculate with, or compare because of their Euclidean geometry, as seen in various references such as Alvarez-Melis & Jaakkola (2018); Gretton et al. (2012); Sejdinovic et al. (2014);

Balasubramanian et al. (2021). Each reproducing kernel Hilbert space thus gives a metric on measures defined by $d(\mu_1, \mu_2) = \|\iota(\mu_1) - \iota(\mu_2)\|$. (Known as the **maximum mean discrepancy**.) Such metrics are also desirable over other metrics on measures such as the Wasserstein metric or other transport-based metrics due to ease of calculation. Note that on a compact space $\Omega$, and a reproducing kernel Hilbert space such that the kernel embedding of measures is injective, we see that the Wasserstein distance is equal to zero if and only if the kernel mean metric is zero– that is, the two distances induce the same metric topology.

Our goal will be to discuss the inverse problem for the kernel embedding of measures– given an element $\nu$ of a reproducing kernel Hilbert space, when does there exist a measure $\mu$ such that $\nu(z) = \int k_\omega(z)\mathrm{d}\mu$. We identify the elements of a reproducing kernel Hilbert space that correspond to measures (and other measure-like elements) supported on a given set $\mathcal{A}$, which provides insight into the data distribution and could potentially facilitate certain various learning tasks.

We will treat several classes of spaces in detail, some of which are mostly of theoretical interest, such as the Hardy space or other spaces of analytic function, but may apply to some applied topics such as learning partial differential equations along the lines of Stepaniants (2023), and others which are more concrete such as Sobelev spaces. Spaces of analytic functions serve as important examples where the kernel embedding of measures is not injective, which serve as interesting examples where complete information may not be recoverable– for example, it is certainly plausible that there are multiple theories describing the same real objects, but which disagree on objects which do not correspond to anything remotely observable in reality, however troubling it may be to our innate longing for one to be right. Irregardless, the problem of whether or not an element of a reproducing kernel Hilbert space remains tractable in terms of duality with the so-called Herglotz functions. Note that, by Caratheodory's theorem, that over an $N$ dimensional reproducing kernel Hilbert space that any such embedded measure can be expressed as a convex combination of at most $N + 1$ kernel functions, which may or may not be supported in the support of the original. We identify the orbits of a dynamical system using the framework of Koopmanism, enabling us to determine when an orbit is contained within a specific compact subset of the domain. Our results offer theoretical foundations and likely practical implications for tasks such as distribution modeling, manifold learning along the lines of Belkin & Niyogi (2003), and anomaly detection (see the survey Chandola et al. (2009).)

## 2 The cone of $\mathcal{A}$-measures

Let $\mathcal{A} \subseteq \Omega$. We define the cone $\mathcal{C}_\mathcal{A}$ of $\mathcal{A}$-**measures over** $\mathcal{H}$ to be the closed cone generated by $\{k_\omega | \omega \in \mathcal{A}\}$. Observe that we can view the membership problem in $\mathcal{C}_\mathcal{A}$ as the inverse problem for the kernel embedding of measures, although in the case where $\mathcal{A}$ is not compact we may not obtain a *bona fide* measure *per se*, but something in their weak closure.

Call a reproducing kernel Hilbert space **uniform** if there exists a point $\omega_0$ such that $k_{\omega_0}(z)$ is positive for every $z \in \Omega$. Note that if $\mathcal{A}$ is compact and the reproducing kernel Hilbert space is uniform, then if $\nu \in \mathcal{C}_\mathcal{A}$, there is a positive measure $\mu$ such that $\langle f, \nu \rangle = \int f(z)\mathrm{d}\mu(z)$.

We note that in a reproducing kernel Hilbert space on a compact set, one can reconstruct the measure by iteratively taking convex combinations with kernel functions, which are the images of point masses under the kernel embedding of measures. Namely, if one picks optimal convex combinations with random kernel functions iteratively, we see that the result converges to the desired kernel mean in countably many steps, . If our set $\mathcal{A}$ is compact, then the space of kernel means is compact, so we indeed get convergence. (Note that if one picks random kernel functions with different distributions, the constructed measures may be different if the problem is not uniquely determined.)

**Theorem 2.1** (Update inequality)**.** *Let $\mathcal{H}$ be a reproducing kernel Hilbert space on $\Omega$. Let $\mathcal{A} \subseteq \Omega$. Let $\mu, \nu$ be $\mathcal{A}$-measures with total variation $1$. If $\mu$ is not equal to $\nu$, there exists $\alpha \in \mathcal{A}$ such that*

$$\|\nu\|^2 > \mathrm{Re}[\langle \mu, \nu \rangle + \mu(\alpha) - \nu(\alpha)].$$

*Moreover, for*

$$t = \min\{\frac{\|\nu\|^2 + \mathrm{Re}[\nu(\alpha) - \langle \mu, \nu \rangle - \mu(\alpha)]}{\|\nu - k_\alpha\|^2}, 1\},$$

*we get that $\|\mu - [(1-t)\nu + tk_\alpha]\|/\|\nu - \mu\|$ is minimized and equal to*

$$\sqrt{1 - \left[\mathrm{Re}\langle \frac{\nu - \mu}{\|\nu - \mu\|}, \frac{\nu - k_\alpha}{\|\nu - k_\alpha\|}\rangle\right]^2}$$

*when $t < 1$ or $\|\mu - k_\alpha\|/\|\mu - \nu\|$ when $t = 1$.*

*Proof.* Note that for any $\gamma \in \mathcal{H}$

$$-\frac{d}{dt}\|\mu - [(1-t)\nu + t\gamma]\|^2|_{t=0} = \mathrm{Re}\langle \nu - \mu, \nu - \gamma\rangle.$$

Thus, if $\gamma$ were an $\mathcal{A}$-measure with total variation 1, we see that there must be some $\alpha$ such that $\|\nu\|^2 + \mathrm{Re}[\nu(\alpha) - \langle\mu, \nu\rangle - \mu(\alpha)] = \mathrm{Re}\langle\nu - \mu, \nu - k_\alpha\rangle > 0$ if $\mathrm{Re}\langle\nu - \mu, \nu - \gamma\rangle > 0$ as $\gamma$ is a limit of convex combinations of kernel functions. (Note that taking $\gamma = \mu$ gives such a positive choice.) Solving for the vertex of the quadratic $\|\mu - [(1-t)\nu + tk_\alpha]\|^2$ gives the desired result. $\square$

Note that, for optimal $\alpha$,

$$\sqrt{1 - \left[\mathrm{Re}\langle \frac{\nu - \mu}{\|\nu - \mu\|}, \frac{\nu - k_\alpha}{\|\nu - k_\alpha\|}\rangle\right]^2} \leq \sqrt{1 - \frac{\|\nu - \mu\|^2}{\|\nu - k_\alpha\|^2}}$$

so we expect a quadratically slow convergence, along the lines of the law of large numbers whenever $\|\mu - k_\alpha\|$ is bounded above. (As the recurrence $t_{n+1} = t_n - \alpha t_n^3$ for $t, \alpha < 1$ goes to 0 quadratically slow. That is, the sequence is $O(\frac{1}{\sqrt{n}})$.) For comparable results on the Wasserstein distance, see Del Barrio et al. (1999).

## 3 Herglotz duality

Given a cone $\mathcal{C}$ in some locally convex topological vector space $\mathcal{V}$, we define the **polar dual** $\mathcal{C}^*$ to the cone of linear functionals with nonnegative real part on $\mathcal{C}$. Note that if $\mathcal{C}$ is a closed cone, the Hahn-Banach theorem implies that $v \in \mathcal{C}$ if and only if, for every $\lambda \in \mathcal{C}^*$, $\mathrm{Re}\lambda(v) \geq 0$.

Taking the cone $\mathcal{C}_\mathcal{A}$, the dual cone is exactly the set of $h \in \mathcal{H}$ such that $\mathrm{Re}\langle h, k_\omega\rangle = \mathrm{Re}h(\omega) \geq 0$ for all $\omega \in \mathcal{A}$. In analogy with classical complex analysis and operator theory, we call the polar dual of $\mathcal{C}_\mathcal{A}$, the cone of $\mathcal{A}$-**Herglotz functions over** $\mathcal{H}$ denoted $\mathcal{H}_\mathcal{A}$.

We call $\mathcal{G} \subseteq \mathcal{H}_\mathcal{A}$ which generates $\mathcal{H}_\mathcal{A}$ as a closed cone an $\mathcal{A}$-**test set.**

Viewing the above discussion (or perhaps more accurately annotated derivation) as a proof, we have the following result.

**Theorem 3.1.** *Let $\mathcal{H}$ be a reproducing kernel Hilbert space on some domain $\Omega$. Let $\nu \in \mathcal{H}$. The following are equivalent:*

1. *$\nu$ is an $\mathcal{A}$-measure over $\mathcal{H}$,*

2. *For every $h \in \mathcal{H}_\mathcal{A}$ we have $\mathrm{Re}\langle\nu, h\rangle \geq 0$,*

3. *Given $\mathcal{G} \subseteq \mathcal{H}_\mathcal{A}$ an $\mathcal{A}$-test set, we have $\mathrm{Re}\langle\nu, h\rangle \geq 0$.*

Note also that the solution to the inverse problem for the kernel embedding of measures is often highly nonunique– for example, for $\mathcal{H}$ a space of analytic functions that the problem for $\mathcal{A}$ compact is equivalent to to the problem for $\partial\mathcal{A}$ by the maximum modulus principle, as they have the same cone of Herglotz functions. Uniqueness problems for the kernel embedding of measures have been analyzed extensively, see the survey Sriperumbudur et al. (2011).

## 4 Over various spaces

### 4.1 The global case over analytic function spaces on the unit disk and the Hardy space in particular

Let $\mathcal{H}$ be a reproducing kernel Hilbert space over the unit disk $\mathbb{D} \subseteq \mathbb{C}$ such that all bounded analytic functions defined on a neighborhood of the unit disk are in our space.

The space of Herglotz functions $\mathcal{H}_{\mathbb{D}}$ is exactly the space of $h \in \mathcal{H}$ such that $\mathrm{Re}\, h \geq 0$. Of course, Herglotz himself classifed the cone of all analytic functions with nonnegative real part Herglotz (1911); Lax (2002)-they are functions of the form:

$$ia + \int_{|\omega|=1} \frac{1 + \overline{\omega}z}{1 - \overline{\omega}z} \mathrm{d}\mu(\omega)$$

where $a$ is some real number and $\mu$ is a finite positive measure on the unit circle. Thus, the cone of relevant Herglotz functions is generated by the elements of the form $\frac{1+\overline{\omega}z}{1-\overline{\omega}z}$ where $|\omega| < 1$ and $\pm i$. which exactly says such functions are a $\mathbb{D}$-test set.

The reproducing kernel for the **Hardy space** $\mathcal{H}^2(\mathbb{D})$ is given by the **Szegő kernel** $k_\omega(z) = \frac{1}{1-\overline{\omega}z}$. Note that $\frac{1+\overline{\omega}z}{1-\overline{\omega}z} = 2k_\omega(z) - 1$. Thus, if $\nu \in \mathcal{C}_{\mathbb{D}}$, we have that $2\nu(z) - \nu(0)$ is a Herglotz function such that $\pm i\nu(0)$ has nonnegative real part. That is, we have the following result.

**Theorem 4.1.** *Let $\nu \in \mathcal{H}^2(\mathbb{D})$. The following are equivalent:*

*1. $\nu$ is a $\mathbb{D}$-measure over $\mathcal{H}^2(\mathbb{D})$,*

*2.*

$$\nu(z) = \frac{\nu(0)}{2} + \int_{|\omega|=1} \frac{1 + \overline{\omega}z}{1 - \overline{\omega}z} \mathrm{d}\mu(\omega)$$

*where $\mu$ is a finite positive measure on the unit circle with total mass $\frac{\nu(0)}{2}$.*

*3. $\nu$ maps $\mathbb{D}$ into the half plane $\{z \in \mathbb{C} | \mathrm{Re}\, z \geq \frac{\nu(0)}{2}\}$.*

Note that the theorem assumes $\nu \in \mathcal{H}^2(\mathbb{D})$.

### 4.2 The global case for entire functions

Suppose $\mathcal{H}$ is a space of entire analytic functions on $\mathbb{C}^n$. Liouville's theorem implies that there are at most only constant Herglotz functions of the from $a + ib$ where $a$ is positive and $b$ is real are in our reproducing kernel Hilbert space. Thus, the cone of $\mathbb{C}^n$-measures over $\mathcal{H}$ is a space of codimension one plus a ray or the whole space.

**Theorem 4.2.** *Suppose $\mathcal{H}$ is a space of entire analytic functions on $\mathbb{C}^n$. Then, either the cone of $\mathbb{C}^n$-measures over $\mathcal{H}$ is all of $\mathcal{H}$ (when there are no constant functions) or a space of codimension one plus a ray.*

*If we additionally assume $\mathcal{H}$ is uniform and thus contains a constant function, we merely need to have the value of $\nu$ at $\omega_0$ to be positive.*

Examples of such spaces include the **Taylor-type spaces**, which arise from taking an entire analytic function $g$ with nonnegative power series coefficients and setting $k_\omega(z) = g(\langle z, w \rangle)$. If $g(0) > 0$, the space is uniform. The classical **Fock space** or **Segal-Bargmann space** (as defined in Bargmann (1961)) is given by $k_\omega(z) = e^{\langle z, w \rangle}$. (Note there are other objects called the Fock space in noncommutative operator theory, see Fock (1932).) Note that the choice of $g$ polynomial gives rise to finite dimensional examples, although one has a paucity of important operators, such as multipliers, composition operators and so on, and thus one must use their truncations.

### 4.3  Real domains and real algebraic geometry

Let $\Omega$ be a subset of $\mathbb{R}^n$. Let $\mathcal{H}$ be a real reproducing kernel Hilbert space. (For example, we could take certain Sobolev spaces, or the space of functions on finitely many points, important for discretizing the problem for applications, among other examples.) In such a case, all the $k_\omega$ must be real-valued functions. Given $\mathcal{A} \subseteq \Omega$, the cone of real-valued Herglotz functions are exactly the nonnegative functions. Thus, the cone of $\mathcal{A}$-measures over $\mathcal{H}$ are exactly $\nu$ such that for every nonnegative $h \in \mathcal{H}$, $\langle \nu, h \rangle \geq 0$.

In Putinar (1993), the following Positivstellensatz, or positivity locus theorem, was obtained, which itself is built of the foundational work of Schmüdgen (1987).

**Theorem 4.3** (Putinar's Positivstellensatz). *Let $g_1, \ldots,$ be real polynomials (either a finite or infinite list) in $n$ variables such that at least one of $G_i = \{z \in \mathbb{R}^n | g_i(z) \geq 0\}$ is compact. Let $g_0 = 1$. Let $\mathcal{A} = \bigcap G_i$. For any polynomial $p$ such that $p > 0$ on $\mathcal{A}$,*

$$p = \sum_{k=1}^{N} q_k^2 g_{i_k}$$

*where $q_k$ are some real polynomials and $i_k$ are some nonnegative integers and $N$ is finite.*

We see the following immediate corollary.

**Corollary 4.4.** *Let $\Omega$ be a subset of $\mathbb{R}^n$. Let $\mathcal{H}$ be a Hilbert space of functions which is closed under complex conjugation such that the polynomials are contained in $\mathcal{H}$ and dense. Let $g_1, \ldots,$ be real polynomials (either a finite or infinite list) in $n$ variables such that at least one of $G_i = \{z \in \mathbb{R}^n | g_i(z) \geq 0\}$ is compact. Let $g_0 = 1$. Let $\mathcal{A} = \bigcap_i \{z \in \mathbb{R}^n | g_i(z) \geq 0\} \subset \Omega$ we have that the collection of polynomials of the form $q^2 g_i$ and $q^2$ are an $\mathcal{A}$-test set.*

Define the $g_i$-**localizing matrix** to be the infinite matrix $A_i = [\langle g_i x^{\alpha+\beta}, \nu \rangle]_{\alpha,\beta}$ where $\alpha, \beta$ run over all multi-indices. We see that $\nu \in \mathcal{C}_\mathcal{A}$ if and only if all the $A_i$ are positive semidefinite. (By which we mean all of its finite principal minors are positive semidefinite.) Thus, we recover exactly the classical conditions on moment sequences as in Schmüdgen (1987); Curto & Fialkow (2005).

### 4.4  Reflexive kernels

Say a real reproducing kernel Hilbert space is **reflexive** if $\mathcal{H}_\Omega \subseteq \mathcal{H}_\Omega^*$ and $k_\omega \in \mathcal{H}_\Omega$. Note, for example, this is the case for the Poisson kernel, $k_w(z) = \frac{1}{1-\overline{w}z} + \frac{1}{1-\overline{z}w} - 1 = \mathrm{Re}\,\frac{1+\overline{w}z}{1-\overline{w}z}$. In such a case, by Theorem 3.1, we have that $\mathcal{H}_\Omega = \mathcal{C}_\Omega$. That is, any Herglotz function on a reflexive space is in the cone generated by the kernels. Note also in a reflexive space that the kernels are positive functions, and the inner product of two Herglotz functions is positive. The conincidental duality can also be used to do finite interpolation.

**Theorem 4.5.** *Let $\mathcal{H}$ be reflexive. Then, $\mathcal{H}_\Omega = \mathcal{C}_\Omega$.*

*Moreover, note that there exists $f \in \overline{\mathcal{H}_\Omega}^{\text{ptwise}}$ such that $f(x_i) = y_i$ if and only if for every sequence of $a_i$ such that $\sum a_i k_{x_i} \in \mathcal{H}_\Omega$, we have that $\sum a_i y_i \geq 0$. (Here $\overline{\mathcal{H}_\Omega}^{\text{ptwise}}$ denotes the closure of the Herglotz functions in the topology of pointwise convergence on $\mathbb{C}^\infty$.)*

Note that, over the disk with the Poisson kernel structure, we obtain exactly the Herglotz representation formula from the above equivalence. Note that, with that structure, the Pick interpolation theorem gives that the above interpolation problem is solvable if and only if there exist $\tilde{y}_i$ such that the matrix $\left[\frac{y_i + y_j + i(\tilde{y}_i - \tilde{y}_j)}{1 - x_i \overline{x_j}}\right]_{i,j}$ is positive semidefinite. It would be most excellent to give a direct proof of the equivalence.

Note that if the kernel functions are nonnegative, then either there is a a pair of Herglotz functions with negative inner product or the space is reflexive. For example, the pluriharmonic Drury-Arveson space (kernel $\mathrm{Re}\,\frac{1+\langle z,w\rangle}{1-\langle z,w\rangle}$ on the Euclidean ball in $\mathbb{C}^d$) in more than one variable is not reflexive, so there must be a pair of Herglotz functions with negative inner product. (We leave the process of finding such a pair to the reader.) In general, the set of harmonic functions on an $\Omega$ with smooth enough boundary are reflexive. Note also that given a space where the kernel functions are nonnegative, one can add some fictional points to the space

$\Omega$ to make the space reflexive. (Corresponding to a maximal cone such that $\langle \nu, \mu \rangle \geq 0$. The utility of such a construction is questionable, especially as the resulting space is unlikely to have nice features, such as a lot of multipliers, but is at least worth mentioning. Such maximal cones are the so-called self-dual cones, which are classified abstractly for general Hilbert spaces in terms of a direct integral theory as described by Penney (1976). Note there are a lot of these, even in finite dimensions– such the nonnegative orthant, right circular cones in $\mathbb{R}^3$ and so on.)

Finally, we point out that a fundamental problem along these lines is given a bounded $\Omega \subseteq \mathbb{C}^d$, when is there a reflexive reproducing kernel Hilbert space on $\Omega$ containing pluriharmonic functions defined on a neighborhood of $\Omega$ as a dense subspace? More generally, one would like to classify all reflexive reproducing kernel Hilbert space structures on a given $\Omega$.

## 5 Koopmanism and an application of inverse problem for the kernel embedding of measures.

Let $\Omega$ be some domain. Let $F : \Omega \to \Omega$. Koopmanism is a popular technique in dynamical systems which (among other things) seeks to find the orbits of a discrete or continuous time dynamical system by linearizing the problem via the theory of composition operators, $K_F h = h \circ F$, which are known as **Koopman operators** in the dynamical context, especially with respect to the dynamic mode decomposition. See Budišić et al. (2012); Brunton et al. (2022). Their adjoints are the Perron-Fronbenius operators, which act on kernels by $P_F k_\omega = k_{F(\omega)}$. The Koopmanism way of doing things (in the discrete time case) should ask when there is a measure supported in some set $\mathcal{A}$ which is invariant under composition, which our context translates to finding eigenvectors $\nu$ such that $P_F \nu = \nu$ and $\nu$ is $\mathcal{A}$-measure over $\mathcal{H}$. Such $\nu$ are called $F$-**invariant** $\mathcal{A}$-measures over $\mathcal{H}$. We say a compact set $\mathcal{A}$ is **Urysohn** on if there exists an $\mathcal{A}$-Herglotz function $h \in \mathcal{H}_\mathcal{A}$ such that $h|_\mathcal{A} = 0$ and $h_{\mathcal{A}^c} > 0$.

**Corollary 5.1.** *Let $\Omega$ be some domain. Let $F : \Omega \to \Omega$ continuous. Let $\mathcal{H}$ be a reproducing kernel Hilbert space on $\Omega$. Let $\mathcal{A} \subseteq \Omega$. The following are equivalent:*

1. *There exists an $F$-invariant $\mathcal{A}$-measure over $\mathcal{H}$*

2. *There is a $\nu$ such that $P_F \nu = \nu$ and $\langle \nu, h \rangle \geq 0$ for every $h$ an $\mathcal{A}$-Herglotz function.*

*Moreover, if $\mathcal{A}$ is Urysohn then such a $\nu$ corresponds to a* bona fide *measure $\mu$ supported on $\mathcal{A}$ and thus $\mathcal{A}$ contains an orbit of $F$.*

*Proof.* It is worth explaining why $\nu$ corresponds to a such a proper measure $\mu$. Pick a measure $\mu_0$ supported on $\mathcal{A}$ representing $\nu$. Now take $\mu_k = \mu_{k-1} \circ F$. Note $\mu_k$ must be supported in $\mathcal{A}$ since it also corresponds to $\nu$. So the support of the measure $\mu = \lim_{N \to \infty} \frac{1}{N} \sum_{k=1}^{N} \mu_k$ is an orbit of $F$. $\qquad\square$

The case of the disk was extremely tractable as we had existing Herglotz representations on the disk. One wonders if one could adopt our framework to be compatiable with Agler models (see Agler & McCarthy (2002)) as adapted to Herglotz functions on the bidisk and study dynamics on the bidisk, with a goal of generalizing existing work such as Sola & Tully-Doyle; Jury & Tsikalas (2023).

**Theorem 5.2** (Operator update inequality)**.** *Let $\Omega$ be some domain. Let $\mathcal{H}$ be a reproducing kernel Hilbert space on $\Omega$. Let $T$ be a bounded linear operator on $\mathcal{H}$. Let $\mathcal{A} \subseteq \Omega$. Let $\nu \in \mathcal{H}$. Suppose there is $\mu \in \mathcal{C}_\mathcal{A}$ such that $T\mu = \mu$. If $T\nu \neq \nu$, there exists a point $\alpha \in \mathcal{A}$ such that the derivative of $f(t) = \|(T-1)((1-t)\nu + tk_\alpha)\|^2$ is negative and, moreover, $f'(t) \leq -2\|(T-1)\nu\|^2$.*

*Proof.* Take $\nu \in \mathcal{H}$ take a random point $\alpha \in \mathcal{A}$ mass $k_\alpha$. Now taking the best linear approximation of $\|(T-1)((1-t)\nu + tk_\alpha)\|^2$ with respect to $t$ at $t = 0$ gives $2\,\mathrm{Re}\langle (T-1)k_\alpha, (T-1)\nu \rangle - 2\|(T-1)\nu\|^2$. Note there is always an $\alpha$ such that $\mathrm{Re}\langle (T-1)k_\alpha, (T-1)\nu \rangle \leq 0$, if there exists such a measure $\mu$ with $(T-1)\mu = 0$. $\qquad\square$

Thus, we see again see about quadratically slow convergence on compact $\mathcal{A}$ if one takes optimal updates by convexly combining with random kernel functions iteratively. Taking $T = P_F$ for some $F : \Omega \to \Omega$ gives the update inequality for the Koopman orbit-detection case. Conceptually, one can view this as a shift in view– instead of iterating to find a steady state, we are picking a point and asking how much mass is missing there, and add it back. Note also that it also statistically feasible to decide when the orbit of every point in $\mathcal{A}$, as if one stops seeing the norm of the defect $\|(P_F - 1)\nu\|$ decrease as in Theorem 5.2, then with more and more confidence we can conclude it will never decrease.

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
