# OpenReview forum: "The inverse problem for kernel means"
_TMLR — Rejected by TMLR_

### Review · Reviewer_wp8q · 2023-06-23

**Summary Of Contributions:**

The paper deals with the inverse problem for kernel mean embedding. That is, given an element in a reproducing kernel Hilbert space, does it correspond to the kernel embedding of a measure? The authors approach this by reformulating the problem as whether an element in an RKHS belong to the so-called "cone of A-measures" over the RKHS. In addition, they discuss an alternative characterisation in terms of its polar dual. They conclude by providing examples of RKHSs and conditions for which an element in the RKHS is a kernel embedding, in addition to providing an application to Koopmanism.

**Audience:**

Yes

**Broader Impact Concerns:**

The paper is purely theoretical and there are no ethical concerns as far as I can tell.

**Claims And Evidence:**

Yes

**Requested Changes:**

- The following objects are not clearly defined, making it hard to understand:
   - (page 2) "We define the cone ... to be the closed clone generated by $\\{k_\omega | \omega \in \mathcal{A}\\}$". More details would be helpful. For example by stating more explicitly $\mathcal{C}_{A} = \\{h \in \mathcal{H} | h(z) = \int_A k_\omega (z) d\mu(\omega) \\}$.
   - (page 3): "We call $\mathcal{G} \subseteq \mathcal{H}_A$ which generates $\mathcal{H}_A$ as a closed cone an $\mathcal{A}$-test set". What does "generate" mean in this context?
- Missing phrase in "We say a compact set $\mathcal{A}$ is Urysohn on [?] if there exists..." (page 6).
- I don't understand the purpose of Theorem 2.1. I believe it is something to do with reconstructing measures from an element in the RKHS (guessing from the context)? This needs to be explained much better and the implications of this result should be stated more clearly.
- Please state where $h$ belongs in the third bullet point of Theorem 3.1.
- The phrase "Note that the theorem assumes $\nu \in \mathcal{H}^2(D)$" (page 4) seems a bit redundant given that the assumption is already included in the statement of the Theorem. Instead of this, it would be more useful for the readers to have a small summary of this subsection (that any element in the Hardy space $\mathcal{H}^2(D)$ is a kernel embedding provided ... is satisfied).
- Section 4.2 -- 4.4 assumes too much from the readers and I believe need more details. For example, in section 4.2, explain in more details why "the cone of $\mathbb{C}^n$-measures over $\mathcal{H}$ is a space of codimension one plus a ray or the whole space". Can you also be more specific about "a space of codimension one plus a ray"?
- Please clarify what is the main purpose of Corollary 5.1. I believe that this is to show that the whole preceding discussion about finding a function in the RKHS corresponding to the kernel mean of a measure can be used to prove the existence of an orbit of $F$ in $\mathcal{A}$? It would be useful to have this intension more clearly expressed and further, would be better if you can make a clearer connection to machine learning. At the moment, this looks more like an application to dynamical systems theory.
- Including some kind of conclusion or discussion section would be useful for the readers.

**Strengths And Weaknesses:**

__Strengths:__
- The introduction is fairly clear compared to the remaining parts, including  a good explanation of kernel embedding and the problem the paper is trying to solve.
- There is a strong display of mathematical knowledge and the insights are quite interesting (especially the way the authors interpret the inverse problem as a membership problem of some cone). However, this can also risk losing the target ML audience.

__Weakness:__
- In general the paper is difficult to comprehend, not only because of the technicalities outside of the standard ML toolbox but also because some parts are (in my opinion) carelessly written. For example, some results assume too much knowledge from the reader, making it seem to appear out of nowhere and some objects are not clearly defined. I believe this can be improved by streamlining the text better, including additional details and providing clearer explanations.
- The applications and significance to machine learning is not very clear (except for the obvious connection with kernel machines, but not sure whether the results in the paper give any useful insights to practical machine learning).

---

### Review · Reviewer_VW2u · 2023-06-26

**Summary Of Contributions:**

In this article, the authors propose to study the inverse problem for RKHSs, that is, whether it is possible to build a measure on the base space whose kernel mean embedding is exactly equal to a given, pre-defined RKHS element. In particular, they discuss how this question can be tackled through Herglotz duality, and then identify several specific cases and kernels (such as Hardy spaces and reflexive kernels) where conditions can be enounced for finding appropriate measures.

**Audience:**

No

**Broader Impact Concerns:**

No concern.

**Claims And Evidence:**

No

**Requested Changes:**

A major rewriting of the article is required to my opinion (see suggestions above).

**Strengths And Weaknesses:**

While the question under study is definitely interesting from a theoretical perspective, I think the article suffers from several major weaknesses:
---The problem is not sufficiently motivated. I understand that the inverse problem for RKHSs is mostly theoretical, but in a contribution to TMLR, there needs to be at least some sort of motivation for applications: why is finding a measure on the data that corresponds to some RKHS point interesting from a practical point of view? Can it help, e.g., to get a better understanding of the structure of RKHSs? Or can it be useful for data visualizaton perhaps?
---The writing can be greatly improved. There are numerous typos and confusing sentences, which makes the article not so easy to follow. In particular, it would really help to add more connections and transitions between the sections as it often feels that the article is jumping from a setting to a drastically different one when going from sections to sections. I would recommend smoothing the flow of the article by explaining why a given section or result naturally leads to the next one. Moreover, some statements include variables that have not been defined (if I understand correctly); e.g., the x_i and y_i in Theorem 4.5.
---It is not clear at all (at least from a non expert point of view) where the contribution actually is. The article looks like a summary of known results and theorems, with the original ones being very incremental (as all can be proved in a few lines). It would greatly help emphasizing what the goal of the article is, whether it aims at gathering a few connected results from the literature, or to bring new ideas to the question.

---

### Review · Reviewer_5Jqi · 2023-07-25

**Summary Of Contributions:**

This is a functional analysis paper on a topic tangential to some challenges within kernel methods in machine learning.  In machine learning, we consider data X or measures from which data is drawn, and consider operating on them with a bi-variate often positive-definite kernel K (e.g., a Gaussian kernel).   Each data point x_i in X with the positive-definite kernel K, encodes a univariate function K(x_i, .), which is an element of a reproducing kernel Hilbert space (RKHS), H_K.  One can then consider operating on these functions as using linear techniques (often implicitly via the kernel trick) to perform various non-linear ML tasks, but retaining convexity etc.  A fundamental task to consider in this context is the kernel mean, which is simply $(1/|X|) \sum_{x_i \in X} K(x_i, .)$.  It can be viewed as a kernel density estimate defined on X.  This all generalizes to continuous measures $\mu$ as well.

Typically in ML the data X is considered in Euclidean space, R^d.  In this case, the reproducing property ensures that for *any* element of H_K there exists some measure $\mu$ (allowing negative weights, or if preferred the difference of two measures $\mu-\nu$) on R^d so the kernel mean of $\mu$ is that element of H_K.
This paper examines more exotic domains, and their corresponding RKHSs, and examine for which elements of those RKHSs does there exist an element of the domain which can generate it.  It turns out unlike the common Euclidean space, this is not always, but can be characterized for a few cases.

**Audience:**

No

**Claims And Evidence:**

Yes

**Requested Changes:**

For me to feel this would be a good fit for TMLR, I would expect to see a description of how the specific results have potential application to machine learning.
I would also hope that the proofs would be written with more details, so more accessible to a typical ML audience.

**Strengths And Weaknesses:**

These results seem neat, but the article is not written for an ML audience, and so is not a good fit for TMLR.  This manifests in two ways:
 1. the connections to ML implications is very tenuous, and only done with some citations in the opening paragraph.  The results on the more exotic spaces are not connected to ML.
 2. the language and detail of the proofs is not written for what an ML audience would expect.  As someone who has written proofs about RKHS in various applications, I found the arguments too terse and with too many elements missing.  In the example in trying to understand Theorem 3.1, there were two (possibly related parts) where I got stuck.
    - In defining the $\mathcal{A}$-test set, I don't understand what "generates" means in "$\mathcal{G} \subseteq \mathcal{H_A}$ which generates $\mathcal{H_A}$".
    - in the third equivalent statement, it uses $\mathcal{G}$ and $h$ but the relation between them is not defined, so its not clear what that third statement means.

I had similar trouble following both the details of the arguments, and the relation to ML for the other sections.

---

### Decision · Action_Editor_7Pdp · 2023-10-27

**Recommendation:** Reject

**Comment:**

Two of the three reviewers gave reject recommendations.  The third reviewer did not provide a recommendation, but with agreement between the other two reviewers (and the AC), this was enough to recommendation rejecting the paper.

There was no author rebuttal on this paper, so the reviewers did not change their original opinion of the paper, which was generally in agreement that the paper is not ready for publication.  In particular, see the comments above on suitability for the TMLR audience and the level of detail of the theoretical results in the paper.  Though kernel methods are certainly important for machine learning, one would like to see an actual direct connection to existing ML methods or some kind of application of the results that would be relevant to an ML audience.  All three reviewers noted clear deficiencies along these two directions.  See the individual reviews for further details, including concerns about the writing of the paper.

**Audience:**

Two of the reviewers felt at least to some degree that the paper fell outside the audience of TMLR.  Reviewer 5Jqi in particular strongly felt that the paper fell outside the ML audience, and reviewer wp8q felt that much of the material in the paper is not easily accessible to an ML audience.  The overall topic could be relevant (namely, reproducing kernel Hilbert spaces are important throughout machine learning), but it seems the paper would need at least a major rewrite before it's ready for publication.

**Claims And Evidence:**

All three reviewers felt the proofs were not sufficiently detailed and had trouble verifying the results.  At the least, the paper would have to be further clarified to improve upon readability.  See the comments from each of the reviewers for further details.